# Adaptable Cryptographic Primitives in Blockchains via Smart Contracts

**Riccardo Longo** *, **Carla Mascia**, **Alessio Meneghetti**, **Giordano Santilli** and **Giovanni Tognolini**

Department of Mathematics, University of Trento, 38123 Trento, Italy; carla.mascia@unitn.it (C.M.); alessio.meneghetti@unitn.it (A.M.); giordano.santilli@unitn.it (G.S.); giovanni.tognolini@unitn.it (G.T.)
* Correspondence: riccardolongomath@gmail.com

**Abstract:** Blockchain-based platforms utilise cryptographic protocols to enforce the correct behaviour of users, as well as to guarantee a sufficient level of protection against malicious adversaries. Cryptography is, however, an ever-evolving discipline, and any breakthrough would have immediate consequences on the security of blockchain-based applications. A possible threat currently under investigation is given by the development of quantum computers, since several wide-adopted cryptographic protocols have been proved to be unsafe against quantum-capable adversaries. In this work, we propose a novel approach for the management of cryptographic primitives in smart-contract-based ledgers, discussing how it fits in both a (partially) permissioned and a fully permissionless setting. The cryptographic protocols are managed in a flexible manner via a set of smart-contracts defined on the ledger itself, in this way the choice of algorithms and parameters can change quickly. Among the advantages of this approach, we remark how it allows designing an adaptive post-quantum-based blockchain that keeps up with ongoing technological advances. In general, the introduction of new features and the application of fixes to a blockchain cause forks in the chain, which may cause major disruptions. The use of smart contracts in blockchain management allows to avoid this problem, dynamically introducing new protocols or deprecating old ones without compromising previous data. The Cryptographic Kernel approach has been adopted by Quadrans, an open-source, public, decentralised smart-contract-based blockchain with a specific focus on the needs of industry, complex supply chains, and IOT devices.

**Keywords:** blockchain; distributed ledger; smart contracts; post-quantum cryptography; digital signatures

## 1. Introduction

Every distributed ledger infrastructure has to adopt a consensus mechanism in order to assure every honest user of the correctness of the recorded data. Among the properties required by blockchains, we place an emphasis upon the immutability of the ledger and the data owner protection.

Regarding the immutability of the ledger, it is clear that there cannot exist multiple versions of the same ledger once the consensus has been achieved, and most consensus protocols rely on cryptographically secure algorithms to guarantee this property. Notable examples are Hashcash and derived proof-of-work protocols [1], which are based on the security properties of cryptographic hash functions, and proof-of-stake protocols [2] often based on digital signature schemes [3].

Data ownership protection is a second, and not less important, property required by distributed ledgers. That is, the owners of some piece of recorded data should have the right to be in charge of its management. To provide an example, only the owner of a digital coin should be able to spend it. On the other hand, the entire network has to be protected against malicious behaviours of data owners: in the context of cryptocurrencies, it should not be possible to spend the same coin twice. Even in this case, cryptographic primitives

are used to enforce a correct behaviour: data ownership and managing rights (within the boundaries of the allowed operations) are usually enforced using digital signature schemes.

In this context, a distributed ledger is safe as long as the adopted cryptographic primitives are secure. However, the security of cryptographic schemes is not absolute and everlasting: advances in the understanding of a scheme may provide evidence of previously unknown vulnerabilities. Any secure public-key cryptographic scheme is based on the intractability of a computationally hard-to-solve mathematical problem; thus, a breakthrough on solving methods for the underlying problem are translated into novel vulnerabilities on the cryptographic scheme, which in turn are directly related to the security of the data stored in a distributed ledger.

### 1.1. Background

The strength and security of a single transaction in a blockchain relies on cryptography, meaning that digital signatures and hash functions should not be compromised. Hash functions are believed to be resistant against both classical and quantum attacks [4], while the same consideration does not apply to the digital signature schemes generally used today.

The most used digital signature schemes are based on the discrete logarithm problem [5]: let $\mathbb{G}$ be a cyclic group, i.e., there exists an element $g \in \mathbb{G}$ that allows any other element $a$ of $\mathbb{G}$ to be written as $a = g^x$ for some integer $x$. The value $x$ is called the Discrete Logarithm of $a$. The problem of determining $x$ once $a$ and $g$ are known is (an instance of) the Discrete Logarithm Problem (DLP). This problem is considered, in the general case, to be intractable in a classical model of computation, namely, we cannot solve the DLP by using a non-quantum computer in a reasonable amount of time [6] for big enough groups. However, over the years several algorithms have been designed to efficiently solve the DLP on particular groups [7]. From a cryptographic point of view, this means that such groups cannot be safely used to design DLP-based schemes.

At the time of writing, in blockchain-based distributed ledgers the most common digital signature schemes are ECDSA [8] and EdDSA [9], whose underlying computationally difficult problem is the ECDLP (the DLP over the group of points of an elliptic curve). If, either due to the development of powerful enough devices or a theoretical breakthrough, ECDLP is no longer be considered difficult, then any application using ECDSA or EdDSA will have to update its protocols to patch their vulnerability. This is particularly important in protocols dealing with sensitive data or financial assets, which are the main use-cases of distributed ledgers. Even if in a classical scenario it is really hard to break a public key scheme based on the Discrete Logarithm Problem (e.g., ECDSA), a quantum algorithm by Peter Shor [10] is able to break such cryptographic primitives in a (relatively) short time. For this reason, current development in quantum computing poses a serious threat also to blockchain-based applications.

To provide a countermeasure against this threat, NIST (National Institute of Standard and Technologies) has proposed a competition to search for new standards of public-key encryption schemes and digital signatures [11], which may be resistant to Shor's quantum attack, and which are therefore called *post-quantum* schemes. We may summarize the main problems on which the most common post-quantum digital signature algorithms rely upon in the following categories:

- Lattice-based: lattices are discrete additive groups on which it is possible to define problems based on the length of the vectors contained in a lattice. These problems have been proved to be post-quantum resistant [12]. Notable examples of this category are CRYSTALS-Dilithium [13] and Falcon [14].
- Multivariate Cryptography: the problem of finding a solution of a system of multivariate polynomial equations over finite fields is believed to be computationally-unfeasible and quantum resistant [15]. Examples of digital signatures of this kind may be found in Rainbow [16] or GeMSS [17].

- Hash-based: since hash functions are believed to be secure against post-quantum attacks [18], some digital signature protocols have been proposed that exploit this feature, such as SPHINCS$^+$ [19].

  A comprehensive list of post-quantum digital signatures may be found in [20].

### 1.2. State of the Art

In the last years, to address the quantum threat on blockchains, some strategies have been presented. For instance, in [21] the authors propose a solution that may lead to a hard fork whenever the digital signature results in being compromised, while in [22] the scheme is improved by performing a soft fork instead of a hard one, when considering a proof-of-work blockchain. Another mechanism is devised in [23], where a simple commit-delay-reveal protocol is presented that enables Bitcoin users to move funds in a secure way from the actual pre-quantum blockchain to a version that implements a post-quantum digital signature scheme. Besides the use of cryptosystems to transition from pre-quantum to post-quantum blockchain, some attempts have been made to design quantum-computing based blockchains, see, e.g., [24–26]. For a complete overview, we refer the reader to [27,28].

The aforementioned solutions risk creating forks to establish a new scheme for digital signatures. We know that the change of an algorithm is not a frequent event during the life of a blockchain, but the main issue of these solutions is that they are not as flexible as one may desire, in terms of usage, space or memory. Therefore it is desirable to adopt a more adaptive approach to the selection and usage of cryptographic primitives and their parameters, favouring some standard choices to optimise efficiency but allowing each user to employ its own choice, in order to balance between security and computational cost, or even to avoid suspicious constructions (i.e., fear of backdoors).

By definition, however, distributed ledgers are designed to be quite rigid in order to enforce immutability, and any change in the content, as well as in what is considered an accepted behaviour of users, has to be accepted by the consensus protocol. Even an update of the list of accepted cryptographic primitives may cause disagreement among the users, thus generating forks of the ledger.

### 1.3. Our Approach

Many ledgers have been developed with the aim of providing methods of performing computations over the ledger itself. For instance, Ethereum [29] is designed as a distributed computer, where any user can define programs, called smart contracts, ask the entire network to run them and record the results on the ledger. The smart contracts themselves are managed through transactions, like any other data in the ledger. Since smart contracts obey the rules of the ledger and are indeed programs capable of running any sort of computation, it would be possible to rely on them to manage the structure of the ledger.

Of course, a ledger whose rules are defined via smart contracts recorded inside the ledger itself should be carefully instantiated, as well as the smart contracts used to define the rules of the ledger. Nevertheless, this kind of structure would allow a flexible governance, solving some of the problems arising from the necessity of updating the rules of the ledger. To provide an example, if the list of acceptable digital signature schemes were to be managed by a specific smart contract, then as soon as a particular scheme is proved vulnerable the nodes could run the script required to mark such scheme as deprecated. Similarly, in case of necessity, more schemes could be added to address specific situations not identified before.

In this paper we propose a useful model for the management of cryptographic primitives adopted in a smart-contract-based ledger. Thus, our approach seems to be the ideal answer to avoid these quantum attacks: a dynamic choice of digital signatures algorithm allows to introduce different solutions to avoid the quantum menace. Moreover, in this way the choice of algorithms can vary quickly and flexibly according to recent developments in cryptanalysis. This choice is further supported by the fact that the research field of

post-quantum cryptography is mostly new, and remarkable studies continue to emerge and highlight substantial weaknesses for these new ciphers (see, e.g., [30]).

### 1.4. Roadmap

In Section 2 we describe the concepts of Cryptographic Kernel (CK) and CK smart contracts: these are the main components in our dynamic update of the blockchain and each of their parts is described in details in Section 2.1. The interactions between users and a CK smart contract for a digital signature scheme in the context of a permissioned blockchain is addressed in the successive sections: in Section 2.2 we describe how a user may consult the CK smart contract to sign a message and how all the other users are able to verify the generated digital signature. In Section 2.3 we discuss how users can add their own parameters sets to the list contained in the CK smart contract, showing the dynamic aspects of our proposal. The update of the tests used to determine the security level of the parameters set is the main topic of Section 2.4, while Section 2.5 concerns the revision of the available algorithms in the list of the CK smart contract. A more concrete example to our adaptive mechanism is suggested in Section 3, where we analyse the Quadrans Blockchain. Finally, in Section 4 we rapidly discuss about the support of CK and CK smart contracts in the context of a permissionless blockchain, and draw some conclusions in Section 5.

## 2. Cryptographic Kernel

To manage a more adaptive design of a blockchain we employ a system of special smart contracts that is called Cryptographic Kernel (CK). We refer to these smart contracts as Cryptographic Kernel Smart Contracts. In particular, they define and encode the available algorithms for each type of primitive, and validate and organise their parameters. In this way, we have a dedicated smart contract for each primitive.

Each CK smart contract contains a list of available algorithms for the primitive. For each algorithm, we gather the following information:

- an implementation reference;
- a list of sets of parameters, achieving different levels of security;
- a list of tests for evaluating the robustness of any set of parameters.

Furthermore, a timestamp is appended to each entry, in order to keep track of all insertions and modifications. We will cover this aspect in great detail later on in the document.

The interactions between the CK smart contracts and the users are regulated as follows:

- all users have read access to the content;
- all users may submit a new set of parameters for an already available algorithm;
- (a pool of) all users may modify the smart contract.

In the following, we focus on the case of a private blockchain, in which only a pool of selected users is in charge of the management and security of the blockchain. We refer to these users as the *Committee*. In Section 4, we will also briefly address the case of a permissionless blockchain, where any user is allowed to update CK smart contracts.

For the sake of simplicity, we reduce ourselves to consider only the smart contract relative to the digital signature primitive.

### 2.1. Structure of the CK Smart Contract

The presence of the cryptographic kernel enables the usage of any algorithm of a non-static pre-approved list, which initially contains a starting selection. Each of these algorithms is uniquely determined by a string identifier, that will be included in any signature in order to specify which digital signature algorithm was used.

Each algorithm is accompanied by a list of parameter sets, a validation suite, and is furthermore equipped with two more fields: "*validated*" and "*deprecated*": once the Committee decides to add a new algorithm to this list, it also applies a timestamp to the *validated* field to keep track of the introduction of the new protocol. However, it may happen that new cryptographic discoveries make an algorithm insecure, therefore the Committee

may decide to flag some algorithms in the CK smart contract as deprecated, in order to warn the users that those algorithms are not recommended anymore. In this case, the Committee adds the timestamp in the *deprecated* field. A schematic view of a CK smart contract is shown in Figure 1.

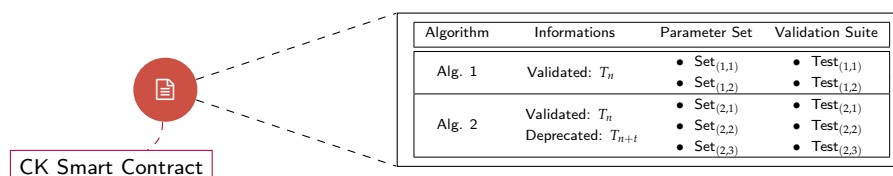

**Figure 1.** The information contained in the CK smart contract: a list of algorithms, parameters sets with their relative security level and timestamps, and the validation suite for each algorithm. Each algorithm has a flag to denote if it has been deprecated.

Since the same version of a given algorithm may slightly differ in some implementation aspects, we also add a reference to recover the specifications of the algorithms presented in the list.

As we said above, the computational power of users varies considerably depending on the resources at their disposal. We may divide the parameters into three different categories, according to the users' needs:

- Basic: suitable for less powerful devices and interaction with legacy applications;
- Intermediate: a middle tier that strengthens the security while balancing efficiency;
- Enhanced: a higher-security level, that is fit for more sensitive and long-term information.

For each level of security, the CK smart contract identifies a set of parameters, identified by the Committee, that is considered as the standard choice.

In addition to the parameters proposed by the Committee, users are allowed to adopt their own, proposing their addition to the CK smart contract. In this way the usage can be furthermore tailored to specific needs. For instance, regarding ECDSA, users can adopt their own elliptic curves, or a different choice of the base point. We will explain this better in Section 2.3.

Each parameter set is accompanied with an assessment that attests the security level that the set clears, and a timestamp that marks when the assessment has been made. If subsequent changes in the security tests cause a variation of this assessment, a new one is appended together with a timestamp of the change. A user should always consider only the latest assessment, except when validating old data in the blockchain, for which we should always consider the information that was available at time of signature creation, so newer assessments should be ignored.

As anticipated, for each of the algorithms the CK smart contract provides also a validation suite, which is used to test the security of the personal parameters proposed by the users. These tests are updated by the Committee as soon as new cryptographic breakthroughs arise and should be constantly revised to guarantee congruent levels of security. In particular, new tests may substitute previous ones, as well as be added to the suite. In any case, after any modification, a timestamp is added in order to keep a record of all the changes in the tests' list and to allow users to check the integrity of the blockchain in its correct temporal context. Whenever the validation suite is modified, all the previously checked parameters have to be tested again with respect to the new version of the tests. Along with the timestamp, the level of security has to be updated according to the new validation results, see Section 2.4.

As an example, consider a first set of parameters which is added at time $T_k$, and suppose that at that time it reaches an intermediate security level. At a later time $T_k + t_1$, a new test could be added to the validations suite and the security of the aforementioned set could degrade to the basic level. Finally, after a possible improvement of the test, assume that, at time $T_k + t_1 + t_2$, the parameter set could no longer be considered secure and in this case its use is deprecated. We refer to Table 1 for a schematic representation of this flow.

**Table 1.** The information contained in the parameter sets and tests fields.

| Parameter Sets | Tests |
|---|---|
| Set 1<br>  • Intermediate: $T_k$<br>  • Basic: $T_k + t_1$<br>  • Deprecated: $T_k + t_1 + t_2$<br>Set 2<br>. . .<br><br>Set $n$ | Test 1<br>  • Added: $T_k$<br><br>Test 2<br>  • Added: $T_k$<br><br>Test 3<br>  • Added: $T_k + t_1$<br>  • Modified: $T_k + t_1 + t_2$ |

An alternative way to manage valid and deprecated algorithms, parameters sets, and tests in the validation suite, is to keep updated two separate lists for each of them: one for the still valid schemes and one for the deprecated ones. Whenever an item is added to one of the list and possibly removed from the other, it is marked with the current timestamp.

*2.2. Signing and Verifying*

In the context of blockchains, a user is often required to sign a transaction or a block. In our setting this operation may be performed according to one of the algorithms described in the CK smart contract: once the users choose the algorithm, first they control that the algorithm is not deprecated, then they set the desired level of security and may pick one of the existing set of parameters or propose a new one. Once the parameters are chosen, they may create their public and private keys pair (or use a pair previously generated for those parameters) and sign the data. Finally, the format of the signed data consists of:

<div align="center">

`algorithm identifier || parameters set || (timestamp) ||signature,`

</div>

where || denotes the concatenation of bytes. The timestamp is optional since the temporal contextualization could be derived otherwise (e.g., by looking at the timestamp of the block where the signature is included). A scheme of this process when the user chooses an existing parameters set is presented in Figure 2, whereas the case in which a new choice of parameters is proposed may be found in Section 2.3.

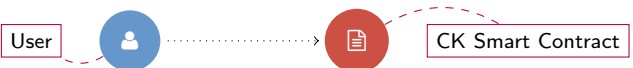

**Figure 2.** Workflow of the signing process: a user consults the CK Smart Contract to decide the algorithm, the level of security and the parameters set.

Once an algorithm and an associated parameter set have been chosen, the user can sign the message and send it to the nodes of the network. At any time, all nodes can easily recognize the signature algorithm by comparing the message header and the CK smart contract. Moreover, they can check the validity and security level of the parameters (and so the signature itself) at the time the signature was created. An exemplary scheme is presented in Figure 3.

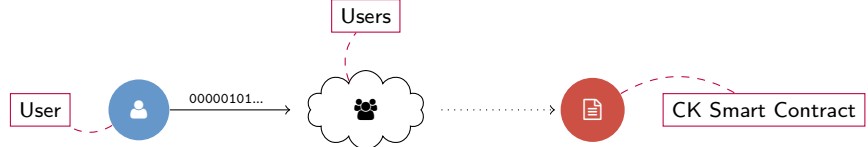

**Figure 3.** Workflow of verification process: on the left a user signs a message and sends it to the others. On the right these other users query the CK smart contract and verify the signature.

### 2.3. Adding New Parameters Sets

One of the main aspects of our system is the possibility for a user to customize the parameters for a given algorithm, making the whole process much more dynamic.

So, as said above, a user who wants to propose their own set of parameters for an algorithm already present in the CK smart contract, may send it to the smart contract itself and test it using the corresponding validation suite (cf. Figure 4).

Now, two possible scenarios are possible:

1. The proposed parameters do not pass the tests needed to guarantee at least a minimum level of security. The CK smart contract returns an error and the user is strongly discouraged to employ those parameters for the creation of digital signatures. Indeed, in this case, other users do not recognize that parameters set as a valid one because it is not contained in the possible choices given by the CK smart contract. Therefore the signature is not considered secure and it is rejected by the community.

2. The parameters pass the mandatory tests provided in the validation suite and are therefore added to the list of permitted parameters for one of the security levels, together with a timestamp. From this moment on, any user may use this new set of parameters.

Anyway, when the validation suite is updated (see Section 2.4), all the sets of parameters authorized in the CK smart contract are tested again for their robustness. Whenever they do not meet the updated requirements, they are flagged as deprecated, time-stamping this decision.

In the second scenario, the storage of parameters that satisfy at least basic security level optimizes the validation effort. Indeed, some tests can require some amount of time and computational power to be performed. In this way, for a given set of parameters, the validation suite runs just once, instead of checking multiple times the same parameters set. On the other hand, the choice of not saving parameters that do not meet the security requirements avoids to consume unnecessary space.

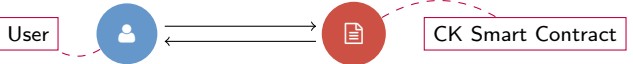

**Figure 4.** Active communication between a user and a given smart contract. The user tries to add a new set of parameter, and the smart contract answers positively or negatively depending on the outcome of the security test.

### 2.4. Update of the Validation Suite

The validation suite and the levels of security are updatable. In fact, we acknowledge both the difficulty of selecting tests which ensure that backdoored parameters will not make it through and the disruption that advances in cryptanalysis may bring. We identify the Committee as the entity responsible of keeping track of the major theoretical discoveries and therefore it is also in charge of the improvements of the validation suite for each algorithm.

Periodically, and whenever new relevant results arise, the Committee decides if it is necessary to fix up the CK smart contract and possibly propose a new list of tests for the algorithms. In this case, the Committee modifies the CK smart contract (time-stamping this modification). The old protocols and parameters in the smart contract are not discarded, so that any user on the blockchain can still declare as valid all the signatures which predate the time of modification (as assessed by the timestamp), avoiding creating a fork in the blockchain.

Each time the smart contract is modified in this way, the tests in the new validation suite for a given algorithm are run again on all the parameters sets. The outcomes of these tests may lead to changes in the levels of security contained in the CK smart contract: parameters that were considered to guarantee a basic, intermediate or enhanced security level may now be vulnerable to some new attacks and need to be classified as achieving a lower level of security. In any case, the results are then confirmed with a timestamp.

To see a possible validation suite for ECDSA, see Example 1.

### 2.5. Adding and Deprecating Algorithms

As research in cryptography is constantly evolving, an important feature for CK is the feasibility to add and deprecate algorithms. Both operations are performed by the Committee, but all users can propose such changes.

As the case of updating the validation suite, the Committee needs to keep up with the last progresses. To add a new algorithm, the Committee has to gather all the information related to the algorithm, as reported in Section 2.1. Therefore, the Committee specifies a bibliographic reference for the algorithm, at least three sets of valid parameters (one for each of the standard security levels) and the validation suite. These data are collected and inserted in the CK smart contract and a timestamp is applied as explained above.

Assume that a new attack is discovered for a digital signature, for instance ECDSA, or that the users (or only the Committee) decide to implement a post-quantum resistant blockchain. In this case, ECDSA has to be considered deprecated in the list of available algorithms in the updated CK smart contract. Anyone should still refer to ECDSA to control that the previous signatures are still valid, checking the timestamps. In Figure 5, we propose an explanatory image in this regard.

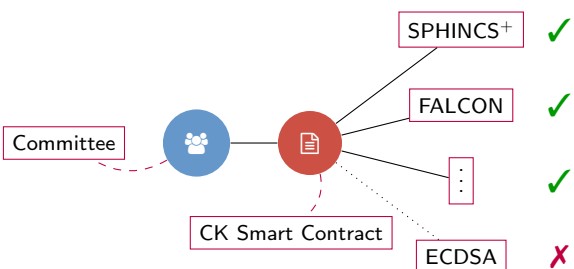

**Figure 5.** The CK smart contract lists all the available algorithms. The committee has decided to deprecate ECDSA, by omitting it in the new version of the CK smart contract.

## 3. Application to Quadrans Blockchain

Quadrans Blockchain [31–33], or simply Quadrans, is an open-source, public, decentralised blockchain. It is devoted to smart contracts with a specific focus on the needs of industry, complex supply chains and IOT devices, with significant designing efforts on the security of this platform and its cryptographic and protocol-related aspects. The development and promotion of Quadrans technology is the main goal of the so-called Quadrans Foundation. It consists of experts, researchers and academics from all over the world, whose aim is to support projects to empower the community and is supported by contributors for the development of an open ecosystem.

Two different currencies are used in Quadrans: Quadrans Tokens and Quadrans Coins. Users possessing the former enjoy special privileges, and they are called TokenHolders.

Quadrans has adopted our Cryptographic Kernel approach, therefore it allows the usage of any algorithm of a non-static pre-approved list, which initially contains a starting selection decided by the Quadrans Foundation and the TokenHolders.

For any cryptographic primitive, the CK smart contract maintains a sub-list of up to 63 standards that are efficiently encoded (see Example 1), from a total of up to $2^{16}$ different algorithms, a limit that they consider over-conservative. The TokenHolders, along with the Quadrans Foundation members, are in charge of updating the CK smart contracts, namely they are responsible for introducing new approved algorithms and removing the obsolete ones.

In the following, we describe how the encoding of the algorithm and parameters works in Quadrans and we present a detailed example concerning the digital signature ECDSA.

The encoding of the choice of the algorithm has been developed so that the majority of cases require just a single byte, that is called the discerning byte. In particular, this

byte encodes the index of the standard algorithm (in the six most significant bits), and the choice of of parameters amongst the standard sets (in the crumb composed by the two least significant bits). The crumbs 01, 10 and 11 correspond to the standard parameters associated to the aforementioned levels of security, namely the basic, the intermediate and the enhanced level; the crumb 00 signals a non-standard choice and therefore it must be followed by 4 additional bytes that specify the index of the chosen parameters in the smart contract list. Similarly, the index 000000 for the standard algorithm signals a non-standard algorithm, so 2 additional bytes that specify the index of the chosen algorithm must follow in this case. Therefore, a null byte indicates a non-standard algorithm with non-standard parameters, so it is followed by 6 bytes: first 2 for the algorithm, then other 4 for the parameters. At this point, the algorithm and parameters are established, so the following bytes can be correctly interpreted referring to the specification defined in the smart contract (e.g., as a public key).

**Example 1.** *Quadrans considers ECDSA as defined in [8], working on elliptic curves in short Weierstrass form defined by the equation:*

$$y^2 = x^3 + ax + b$$

*over a field $\mathbb{F}_p$ of prime order $p$. The parameters of ECDSA are the prime $p$ and the values $a, b$ (each of n bits) that define the curve E, plus a base point $\mathcal{B} = (\mathcal{B}_x, \mathcal{B}_y)$ of prime order $q$. For the sake of completeness, we specify here the key generation algorithm, the signature algorithm and the verification algorithm of ECDSA.*

*Key Generation*

*Create a key-pair performing the following steps:*
- *The private key is an integer d selected uniformly at random in the interval $[1, q-1]$.*
- *The public key is the elliptic curve point $\mathcal{Q} = d\mathcal{B}$.*

*Signing*

*Given a key-pair $(d, \mathcal{Q})$ and a message digest M, compute the signature performing the following steps:*
- *Pick an integer k uniformly at random in the interval $[1, q-1]$.*
- *Compute the point $\mathcal{R} = (\mathcal{R}_x, \mathcal{R}_y) = k^{-1}\mathcal{B}$.*
- *Interpret the message digest M as an integer e.*
- *Compute $s = k(e + rd)$, where $r = \mathcal{R}_x$.*
- *The signature is the pair $(r, s)$.*

*Verification*

*To verify a signature $(r, s)$ on a message digest M perform the following steps:*
- *Check that $r, s \in [1, q-1]$.*
- *Interpret the message digest M as an integer e.*
- *Compute $u_1 = es^{-1} \bmod q$ and $u_2 = rs^{-1} \bmod q$.*
- *Compute the point $\mathcal{U} = (\mathcal{U}_x, \mathcal{U}_y) = u_1\mathcal{B} + u_2\mathcal{Q}$.*
- *Check that $r \equiv \mathcal{U}_x \bmod q$.*

In Quadrans, ECDSA is the first standard algorithm for Digital Signatures, so the value *000001* of the six most significant bits of the discerning byte currently symbolises ECDSA.

In defining and encoding ECDSA parameters, Quadrans tries to minimize length and maximize efficiency. In this regard, it adopts point compression, so public keys can be encoded using only the x coordinate of the corresponding point, from which the y can be derived, assuming its parity or quadratic residuosity (depending on p). Another efficiency-enhancing strategy exploits the fact that the values a and b can be significantly shorter than p, so a variable-length encoding (inspired by the Recursive Length Prefix encoding used in Ethereum [34]) is used.

*Following these principles, a public key is encoded (just like the base point in the parameters) as a string of bytes that represent the x-coordinate of the point (note that the length of this string can be deduced from the parameters).*

*A set of ECDSA parameters is validated before insertion in the smart contract list performing the following tests (motivation for the tests can be found in [35]):*

1. *$p$ must be prime;*
2. *$q$ must be a prime different from $p$;*
3. *$a, b, p$ must define an elliptic curve, i.e., a non-singular cubic;*
4. *the point $\mathcal{B} = (\mathcal{B}_x, \mathcal{B}_y)$ must belong to the elliptic curve and be different from the neutral element $\mathcal{O}$ of the group;*
5. *$q$ must be the order of $\mathcal{B}$ (it is sufficient to check that $q \cdot \mathcal{B} = \mathcal{O}$);*
6. *security against Pollard's rho algorithm: $\log_2\left(\sqrt{\frac{\pi}{4}q}\right) =: \rho > 78$;*
7. *security against transfers: the order of $p$ in $\mathbb{F}_q^*$, called* embedding degree, *must be greater than $\frac{q-1}{78}$, where $\mathbb{F}_q^*$ denotes the multiplicative group of non-zero elements of $\mathbb{F}_q$;*
8. *optional test: the complex-multiplication field discriminant should be larger than $2^{78}$.*

*The (optional) discriminant test requires the factorization of a big integer, and this is a very expensive operation, however verifying a factorization and the primality of an integer is relatively cheap, so Quadrans uses a support smart-contract that stores a list of prime numbers that can be used to streamline security checks. In particular, when submitting a set of parameters to be checked a user should also add to the list of primes all prime divisors of $t^2 - 4p$ (where t is the trace), obviously the list should avoid duplicates.*

*Upon insertion in the list each set is accompanied by an extra byte at the beginning that sums up the security assessment according to the tests. Initially, only the meaning of few values is defined, leaving ample space for future developments that may expand the tests to keep up with cutting-edge cryptanalysis. At the moment, the security level is determined by the value of $\rho$ as computed in the test 6: the basic level requires $\rho > 125$ (the standard set for this level is Curve secp256k1 [36]), the intermediate level requires $\rho > 185$ (the standard set for this level is Curve brainpoolP384t1 [37]), the enhanced level requires $\rho > 250$ (the standard set for this level is Curve brainpoolP512r1 [37]).*

*Suppose that the following string is the hexadecimal representation of a public key:*

$$\underset{\textit{discerning byte}}{\underbrace{\texttt{05}}} \; \overbrace{\texttt{82006e9398a6986eda61fe91674c3a108c399475bf1e738f19dfc2db11db1d28}}^{\textit{public key}}$$

*The first byte (`05`) identifies the scheme and therefore how to process the rest. Its binary representation is:*

<div align="center">

`000001 01`

</div>

*The first (and most significant) six bits identify the first standard algorithm defined in the digital signature algorithms CK smart contract (i.e., ECDSA), the last (and least significant) two bits say that the basic-level standard is used (referring to the ECDSA parameters smart contract, i.e., the curve secp256k1). This means that what follows the first byte is the encoding of the public key, since the curve parameters are established and known. Note also that the parameters define the length of the encoding, so the parser knows how many bytes to read. ECDSA's public key is a point of an elliptic curve, and the parameters used imply the quadratic residuosity of the second coordinate, so the remaining sixteen bytes of the public key are the first coordinate of the point.*

## 4. Future Directions

In our model, we focused on not-entirely permissionless blockchains, so that we could rely on the presence of a sort of authority in charge of the safe management of the cryptographic kernel. Unless we assume that the authority is a selected pool of users with more duties and privileges, this choice corresponds to the necessity of relying on a trusted third party. While this feature could be acceptable in certain use-cases, in the general

context of blockchains one could yearn for a system where all users contribute equally to the maintenance of the ledger.

To extend our model to a public and permissionless blockchain, the first, naive, approach would be to give to each and every user the right to freely update any CK smart contract. Indeed, if not managed by an authority, the cryptographic kernel is nothing more than a set of smart contracts like any other. Its sensitive role however makes this set of smart contracts a target for breaching the security of the entire system.

Imagine that our Cryptographic Kernel were to be adopted in a standard PoW Bitcoin-like blockchain. If any user were allowed to modify the CK smart contracts, then by adding vulnerable cryptographic protocols and by deprecating the secure ones, a malicious user could compromise the entire ledger. If a malicious user is capable of winning a PoW competition even just once, then it could add to the ledger an update of the CK smart contracts and convince honest users to adopt a vulnerable digital signature scheme. From that moment onward, the malicious user could therefore compromise the fooled users. In the worst-case scenario, in which the majority of users are not capable of discerning between vulnerable and secure protocols, this attack could be devastating, otherwise a fork could revert the malicious modifications, but nevertheless it would cause serious discomforts if the injected vulnerabilities are not spotted before they are used by unsuspecting users. For these reasons we believe that suitable countermeasures should be considered, being aware that some may not be directly obtainable in a permissionless ledger.

A proposal that strives to reconcile the need of expert evaluation of algorithms and parameters, with a maximum freedom when proposing and adopting them, relies on a concept of reputation that resembles the web of trust. In this approach the warden-like role of the committee is taken over by multiple experts which endorse or disapprove algorithms, parameters, and possibly even other experts. The "expert" status should be awarded to users with sufficient cryptographic competences, but their appointment could be entirely democratic. For example, some reputed researchers could candidate themselves to become experts: by showing their academic credentials they convince users to back them (e.g., by signing some sort of trust declaration on chain). Once they have sufficient following, they can officially evaluate additions to the cryptographic kernel. Within this construction, users can then use algorithms and parameters endorsed by widely supported experts with good confidence that the other users will deem them safe (thus they will not reject them). Note that an implementation of this approach should however be carefully studied in the context of signaling games [38,39], analyzing how effectively the experts can convey their evaluations of primitives, and the users can discriminate the competence of the experts.

Another proposal to extend the model of Cryptographic Kernel to the case of public blockchains is to adopt a multi-level approach in the management of the ledger. For example, the Cryptographic Kernel could belong to a sub-blockchain managed by a committee of nodes elected among the entire pool of users, which are temporarily in charge of the CK smart contracts. In this example, any update of content of the Cryptographic Kernel would be subject to the evaluation of this committee, which has to reach an internal consensus in order to validate the update. If carefully designed (e.g., by adopting a selection of the committee based on cryptographically secure pseudo-random functions and a BFT consensus mechanism), it should be possible to prove that malicious users can successfully attack only with negligible probability. Of course, any model for the case of a permissionless blockchain should be designed by taking into account the specific purpose of the platform.

Future works will therefore have to consider specific (public) distributed ledgers, to assess the possibility of including a Cryptographic Kernel and achieving the desired flexibility and security offered by our model.

## 5. Conclusions

We introduced the concept of Cryptographic Kernel (CK) for blockchain-based applications. We have shown how CK enables dynamic and flexible management of the cryptographic primitives employed by users through smart contracts. The use of a custom

set of permitted algorithms with different levels of security allows a more active management of the security of the blockchains, protecting almost immediately against possible attacks that may eventually be discovered. This new approach gives back control to users, letting them choose the most suitable algorithms according to their computational resources and needs. Moreover, the adoption of the CK may be the key to designing an adaptive post-quantum-based blockchain that keeps up with ongoing technological advances, solving one of the major problems that blockchains will have to face in the near future.

**Author Contributions:** Conceptualization, R.L., C.M., A.M., G.S. and G.T.; Formal analysis, R.L., C.M., A.M., G.S. and G.T.; Investigation, R.L., C.M., A.M., G.S. and G.T.; Methodology, R.L., C.M., A.M., G.S. and G.T.; Supervision, R.L., C.M., A.M. and G.S.; Visualization, C.M., A.M., G.S. and G.T.; Writing—original draft, R.L., C.M., A.M., G.S. and G.T.; Writing—review & editing, R.L., C.M., A.M., G.S. and G.T. All authors have contributed equally to this work. All authors have read and agreed to the published version of the manuscript.

**Funding:** The publication was created with the co-financing of the European Union—FSE-REACT-EU, PON Research and Innovation 2014–2020 DM1062/2021. The APC was funded by Quadrans Foundation.

**Acknowledgments:** The authors are members of the INdAM Research group GNSAGA.

**Conflicts of Interest:** The authors declare no conflict of interest.

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
