# Peer review of "Adaptable Cryptographic Primitives in Blockchains via Smart Contracts"

_cryptography, doi:10.3390/cryptography6030032_

Round 1
Reviewer 1 Report
Please revised the paper in the following parts:
1) Please extend abstract to at least 250 words. Currently it is too short to serve the purpose.
2) Extend Introduction as well, paragraphs are too short.
3) What are the key contributions of this paper. Explain in Introduction.
4) Please explain the significance of your work, and discus how this work is different from the existing knowledge?
5) How this work validated? Explain please.
Author Response
1) Please extend abstract to at least 250 words. Currently it is too short to serve the purpose.
We have extended our abstract to 250 words, trying to make it more suitable to serve the purpose of the document.
2) Extend Introduction as well, paragraphs are too short.
The introduction is divided in subsections as suggested by the reviewers after the first submission. As a whole the introduction is three pages long.
3) What are the key contributions of this paper. Explain in Introduction.
In the sub-section 1.3 of the Introduction titled "Our approach", we expose our main contributions.
4) Please explain the significance of your work, and discus how this work is different from the existing knowledge?
The significance of our work, together with the discussion about how this work is different from the existing knowledge, are presented in the two sub-sections of the Introduction 1.2 "State of the Art" and 1.3 "Our approach".
5) How this work validated? Explain please.
The work has been developed on a theoretical level, and there is no implementation yet. Therefore we validated our results from an abstract point of view showing how we reached our goal of dynamic management of cryptographic primitives.
To support our statements, we also provided an example of how our approach is being adopted by a real-world platform (section 3, example 1).
Reviewer 2 Report
In this paper, a Cryptographic Kernel which is blockchain based strategy is suggested. The technique can be further improved by addressing following concerns:
1. Abstract should be modified by considering the discussion of the proposed work and the experimental results.
2. The objective and research issue should be discussed in a separate section.
3.How the proposed model is useful for the management of cryptographic primitives adopted in a smart-contract-based ledger is missing.
4. Introduction and background details can be further strengthened by discussing recent works
5. Example 1 discusses ECDSA without proper background details of it. Authors are suggested to present a clear illustration for clear understanding.
6. Validation of the cryptographic kernel should be performed with proper discussion and illustration.
7. Results and discussion section is missing. Authors should establish a comparative analysis with other existing works.
Author Response
1. Abstract should be modified by considering the discussion of the proposed work and the experimental results.
We have extended our abstract to 250 words, including some discussion of the proposed solution and its application.
2. The objective and research issue should be discussed in a separate section.
The research issue is introduced in sub-section 1.2 of the Introduction, while in sub-section 1.3 we expose our main contributions.
3. How the proposed model is useful for the management of cryptographic primitives adopted in a smart-contract-based ledger is missing.
This aspect has been remarked in the abstract, in sub-section 1.3, in section 5 and is the core of section 2.
4. Introduction and background details can be further strengthened by discussing recent works
In the subsection 1.2 titled "State of the Art" we discuss recent works, with an emphasis on how the current solutions do not fully address the problem at the heart of our paper.
5. Example 1 discusses ECDSA without proper background details of it. Authors are suggested to present a clear illustration for clear understanding.
We have extended the example 1 of section 3 in order to cover the basics of ECDSA.
6. Validation of the cryptographic kernel should be performed with proper discussion and illustration.
The work has been developed on a theoretical level, and there is no implementation yet. Therefore we validated our results from an abstract point of view showing how we reached our goal of dynamic management of cryptographic primitives.
To support our statements, we also provided an example of how our approach is being adopted by a real-world platform (section 3, example 1).
7. Results and discussion section is missing. Authors should establish a comparative analysis with other existing works.
As showed in sub-section 1.2 ("State of the Art"), existing works do not properly address the problem we aim to solve with our proposed solution, therefore it does not seem possible to make a meaningful comparative analysis.
In section 5 ("Conclusions") we discuss the results obtained and emphasize the novelty of our approach and how it improves the existing solutions.
Round 2
Reviewer 1 Report
The paper is improved after revision. So my decision is accept in present form.
Reviewer 2 Report
The authors have successfully addressed mu concerns
This manuscript is a resubmission of an earlier submission. The following is a list of the peer review reports and author responses from that submission.
Round 1
Reviewer 1 Report
Please revise the paper in the following parts:
1) Abstract is too short to serve the purpose. Extend it please.
2) Please refine presentation of key contributions in Introduction.
3) How your work is different from existing research? Please create a research gap (in an Related Works) to justify the significance of your work.
4) How this work is validated? Please incorporate "Results and Discussion" section with simulation details and supporting results.
5) Conclusion section is missing. Support your conclusion with key results, and research insights.
Reviewer 2 Report
The manuscript does not follow the structure of a scientific paper.
It reads like a survey?
The problem is not well articulated.